# Pruning compact ConvNets for Efficient Inference

## Abstract

Neural network pruning is frequently used to compress over-parameterized networks by large amounts, while incurring only marginal drops in generalization performance. However, the impact of pruning on networks that have been highly optimized for efficient inference has not received the same level of attention. In this paper, we analyze the effect of pruning for computer vision, and study state-of-the-art ConvNets, such as the FBNetV3 family of models. We show that model pruning approaches can be used to further optimize networks trained through NAS (Neural Architecture Search). The resulting family of pruned models can consistently obtain *better* performance than existing FBNetV3 models at the same level of computation, and thus provide state-of-the-art results when trading off between computational complexity and generalization performance on the ImageNet benchmark. In addition to better generalization performance, we also demonstrate that when limited computation resources are available, pruning FBNetV3 models incur only a fraction of GPU-hours involved in running a full-scale NAS.

## 1 Introduction

Neural networks frequently suffer from the problem of *over-parameterization*, such that the model can be compressed by a large factor to drastically reduce memory footprint, computation as well as energy consumption while maintaining similar performance. This is especially pronounced for models for computer vision (Simonyan & Zisserman, 2014), speech recognition (Pratap et al., 2020) and large text understanding models such as BERT (Devlin et al., 2018). The improvements obtained from intelligently reducing the number of model parameters has several benefits, such as reduction in datacenter power consumption, faster inference and reduced memory footprint on edge devices such as mobile phones which also enable decentralized techniques ex. federated learning (Kairouz et al., 2019).

There are several techniques to reduce model size while maintaining similar generalization performance, such as model quantization (Polino et al., 2018), NAS (Neural Architecture Search) (Elsken et al., 2019) and model distillation through teacher-student networks (Gou et al., 2021). For the scope of this paper, we consider pruning as a technique to remove trainable weights in the network, and save on computation costs for the FBNet family of models. The motivations for this are two-fold. Firstly, state-of-the-art models such as FBNet (Wu et al., 2019) already adopt the best practices in the area of efficient hardware-aware design of convolutional neural network based models, and are widely used across different vision tasks. This makes them suitable baselines to understand whether pruning can offer any performance gain over their already optimized behavior. While there has been limited work on pruning for efficient convolution network models they investigate older architectures such as EfficientNet and MobileNet (Aflalo et al., 2020) or integrate pruning into expensive techniques such as joint prune-and-architecture search (Wang et al., 2020).

For each of the constituent models of the FBNetV3 family (FBNetV3A, FBNetV3B,..., FBNetV3G) we reduce the number of parameters using two pruning based approaches: (1) *Global magnitude-based pruning*: Starting with the pre-trained model, we prune all weights whose magnitude is below a threshold chosen in order to achieve a target number of FLOPs for the pruned model; (2) *Uniform magnitude-based pruning*: Starting with the pre-trained model, we prune weights in each layer whose magnitude is below a level-specific threshold in order to yield a pruned model achieving a target number of FLOPs with the same sparsity in each layer. After either pruning method is applied, we

fine-tune the pruned model for a certain number of epochs until convergence is reached. Within the scope of our study in this paper, we are mostly interested in the following research questions:

- **RQ1**: Pruning to improve computation vs. performance tradeoff. Can a model obtained by pruning a larger FBNetV3 model **M1** (optimized using NAS) achieve higher generalization performance than a smaller FBNetV3 model **M2** when the pruned model has the same number of FLOPs as **M2**?

- **RQ2**: Pruning as an efficient paradigm. When a larger FBNetV3 model **M1** is available and computational resources are limited, is pruning a faster and less computationally expensive approach to obtain a model with higher accuracy at a desired computation level (FLOPs) than running a full-fledged architecture search?

*Pruning to improve computation vs. performance tradeoff (RQ1).* There have been recent research advances in the area of building hardware-aware efficient models (Deng et al., 2020). These can provide good generalization performance while adhering to constraints on memory, inference latency and battery power, which are often dictated by the hardware environment where inference happens. Experiments described in existing work on efficient vision models such as ChamNet (Dai et al., 2019), MobileNet (Howard et al., 2017), EfficientNet (Tan & Le, 2019) and FBNetV2 (Wan et al., 2020) have shown that it is possible to achieve even higher performances on standard image recognition tasks such as ImageNet (Deng et al., 2009) at a certain level of FLOPs. However the efficient design of these models does not solve the over-parameterization problem completely, and none of these approaches study how model pruning can be performed to obtain even better trade-offs between computation and model accuracy. This paper is the first of its kind to understand how we can improve on the state-of-the-art in this problem space.

*Pruning as an efficient paradigm (RQ2).* In addition to achieving state-of-the-art performance with reduced FLOPs, we are also interested in understanding how such pruned models can be obtained *inexpensively* with limited resources that are generally available to a machine learning practitioner who has access to existing optimized models but limited computing resources. For example, the FBNetV3 models are freely available through Facebook's Mobile Model Zoo[1], while EfficientNet models can be obtained at GitHub[2]. While the techniques needed to obtain computation- and latency-friendly models have been democratized through open-sourcing the source code as well as the models themselves, fully applying these techniques necessitates costly operations such as finding an optimal network topology through meta-learning approaches (You et al., 2020) and search algorithms such as Genetic Algorithms (GAs) (Goldberg & Deb, 1991).

Given the high-degree of intractability of this problem, expensive computational resources are often needed in this case, easily exceeding the budget available to a university research laboratory or an angel-stage startup (Zoph & Le, 2016). When a starting model is already available, for example through open-sourcing, the best option would be to perform a cheap modification of the model to fit a certain target FLOPs/latency requirement. In this paper we have compared the NAS approaches for training FBNetV3 models with our pruning techniques on a computational complexity metric (GPU-hours) to effectively answer **RQ2**.

*Benchmark results.* In addition to experimental outcomes for answering **RQ1** and **RQ2**, we also benchmark pruned FBNetV3 models using available open-sourced quantized sparse kernels and conduct ablation studies to obtain additional insights into pruning performance. These results augment our main observations and demonstrate that with existing hardware support, it is possible to deploy pruned cutting-edge computer vision models with practical latency reductions and improve further beyond the performance vs. FLOPs trade-off.

We conduct our experiments on ImageNet, which is an object-recognition task on a large training dataset of 1.2 million images. We show that computationally less intensive techniques such as uniform and global magnitude-based pruning of larger FBNetV3 models can yield higher test accuracies than small models while having the same number of FLOPs. Given a target computation budget for an efficient model, we show that it is more practically advantageous (both in terms of performance and running time) to simply prune the larger model than run a neural architecture search to find the target model from scratch.

---

[1]FBNetV3 models available here `http://https://github.com/facebookresearch/mobile_cv/model_zoo/models/model_info/fbnet_v2/model_info_fbnet_v3.json`

[2]EfficientNet models available here `https://github.com/mingxingtan/efficientnet`

The technique we have employed for pruning (unstructured sparsity) is already tried and tested, however our novelty lies in studying whether efficient image recognition models such as FBNetV3 can be optimized further to improve on the FLOPs-accuracy curve, and the contributions are two fold : (1) FBNets are themselves state-of-the-art in efficient vision models and we achieve better accuracy-FLOPs tradeoff over these models and (2) from the standpoint of computational overhead, we significantly reduce the amount of GPU hours required to obtain such models. Pruning a publicly available NAS optimized model incurs ≈4x less GPU hours to achieve a target FLOPs level, compared to training a full-fledged NAS to obtain a model which has less accuracy at the same FLOPs level.

*Paper organization.* The remainder of this paper is organized as follows. In Section 2, we describe related work in the area of efficient vision model design and also provide an introduction to different pruning techniques. In Section 3, we discuss our experimental setup, including a description of the baseline models and the *global* and *uniform* pruning approaches we have employed. Section 4 describes our main findings and we conclude the paper in Section 5.

## 2 Related Work

We discuss related literature in the areas of *computationally efficient vision models* and *model pruning*. Within the scope of our work, we mainly focus on inference efficiency of models in contrast to training efficiency.

*Computationally efficient vision models:* Neural networks for computer vision are generally characterized by convolutional layers and fully-connected layers, along with blocks such as residual or skip connections. This makes such networks resource intensive in terms of FLOPs, which affects the memory storage and power consumed, and also leads to increased latency. It is of paramount importance to design more efficient networks which can provide higher performance for the same FLOPs or latency level, or even to optimize them appropriately to provide the same performance at reduced FLOPs/latency. This can be performed either through the design of new simplified layers, for example in deep residual learning (He et al., 2016) or though explicit model compression as in weight quantization (Polino et al., 2018). Extremely deep networks for image recognition often suffer from not only high complexity and inference latency, but also from the issue of *vanishing gradients* (Pascanu et al., 2013). This was addressed through deep residual networks which effectively simplified network design through skip-connections. MobileNets (Howard et al., 2017) are one of the earlier approaches to building small low-latency networks by using depthwise separable convolutions with two parameters, *width* and *resolution* multipliers. They demonstrate the effectiveness of MobileNets across different vision tasks, such as face embeddings and object detection. MobileNetV2 (Sandler et al., 2018) extends MobileNets by utilizing inverted residual filter structures and linear bottlenecks, obtaining improvements on state-of-the-art models both in terms of accuracy and computational complexity. ShuffleNets (Zhang et al., 2018) propose dedicated residual units where $1 \times 1$ convolutions are replaced with pointwise group convolutions and channel shuffling reducing FLOPs computations.

More recently, the focus on building efficient neural network models has shifted to techniques that treat the design of efficient networks as a search problem, falling under the umbrella of Neural Architecture Search (NAS). EfficientNets (Tan & Le, 2019) propose a novel scaling method which adjusts the network's length, width, and resolution to optimize performance subject to target memory and FLOPs constraints. They also define a novel baseline that is optimized by a multi-objective neural architecture search. The FBNet collections of models—FBNet (Wu et al., 2019), FBNetV2 (Wan et al., 2020) and FBNetV3 (Dai et al., 2021)—employ neural architecture search to obtain highly-optimized models that improve on the state-of-the-art for different visual understanding tasks. FBNet frames the architecture search as a differentiable meta-learning problem with gradient based techniques, namely *DNAS*—Differentiable Neural Architecture Search—by Wu et al. (2019), and avoids selecting the optimized model over a discrete set. The subsequent entry in this collection, FBNetV2, expands the search space over conventional DNAS, and employs a masking scheme to maintain the same level of computational complexity while searching over this expanded space. FBNetV3 further improves on the state-of-the-art by employing Neural Architecture Recipe Search (NARS) and searching over the space of not only architectures, but also corresponding recipes (which are generally hyper-parameters). In this paper, we consider FBNetV3 models as our baselines as they are state-of-the-art. We are interested in understanding if they are overparameterized and evaluate how much model pruning can improve performance at a certain FLOPs level over the state-of-the-art in this family of models.

*Model Pruning:* Modern neural networks, particularly those processing complex sensory inputs (such as speech, vision and language) for perception applications, are often over-parameterized. It is only to be expected that we should be able to compress such networks significantly to maintain the same level of performance at decreased level of computation (fewer weights and reduced FLOPs), memory footprint and power consumption. Foundational efforts in this space include the *Optimal Brain Surgeon* (Hassibi & Stork, 1993) and *Optimal Brain Damage* (LeCun et al., 1990). Recently the idea of network pruning has been formalized through the lottery ticket hypothesis (Frankle & Carbin, 2018), which claims that randomly initialized, feed-forward networks have winning sub-networks that perform just as well as the original network on an unseen test dataset. Model pruning is generally of two types: unstructured and structured pruning. Unstructured pruning, as the name suggests, doesn't adhere to any structure and prunes neurons based on chosen criteria (such as magnitude). This has the advantage of providing higher performance, but is difficult to implement in hardware, as it needs dedicated support for efficient sparse matrix multiplications. Meanwhile, structured pruning is the practice of removing entire groups of neurons (e.g., blocks within the weight matrix, or channels in convolutional neural networks). This is easy to implement without dedicated hardware support, but has the issue of lower generalization performance than unstructured pruning (Yao et al., 2019). In the literature, there have also been several studies, for example investigating whether rewinding (training from scratch with a fixed mask) can perform just as well as the fine-tuning on top of the original unpruned network (Renda et al., 2020). Blalock et al. (2020) provide an overview survey of recent advances and open problems in neural network pruning.

In the research area of designing efficient networks for computer vision, there has not been much focus on understanding how pruning can be applied to the current generation of models. Most literature on pruning is based on older networks such as VGGNet, ResNet (He et al., 2016), and MobileNet (Sandler et al., 2018). Our work improves upon these existing studies by understanding how pruning can improve the FLOPs-accuracy tradeoff over existing state-of-the-art networks.

## 3 PRUNING TECHNIQUES AND SETUP

In this section, we describe the main components of our techniques and experimental setup, including *Baseline Models*, *Pruning Techniques*, *Latency Measurement* and *Metrics*. We have mainly used standard splits of the ImageNet dataset, further details are in Section A.1 of the appendix.

### 3.1 BASELINE MODELS

Dai et al. (2020) address the previous limitations of NAS-based architecture search where these approaches can only search over architectures given a training recipe (set of hyperparameters), and thus cannot optimize over both. As described in Section 2, the most recent state-of-the-art models are based on NARS (Neural Architecture-Recipe Search), which we select as baseline models. Table 3 lists the accuracy of FBNetV3 models (Dai et al., 2021) on the ImageNet classification task, along with the number of model parameters and computation complexity in terms of FLOPs.

Each baseline model consists of multiple IRF (Inverted Residual Filter) blocks, which contain convolutional layers of different kernel sizes. For our experiments, we are mostly interested in $1{\times}1$ convolutions as potentially prunable, since within each FBNetV3 model, the $1{\times}1$ convolution layers constitute >80% of total model FLOPs for all models in the family, and the open-sourced sparsity kernel support we use for latency benchmarking is available only for fully connected layers. A $1{\times}1$ convolution can be transformed into an equivalent fully connected layer with a few tensor reshape operations without any significant loss of performance or latency.

For each initial and target FBNetV3 model $X$ and $Y$, where $X$ is larger than $Y$, we prune $X$ to a *sparsity level* of $S$ so that the FLOP count is the same as for $Y$. The number of FLOPs consumed by a linear layer of sparsity $S$ is proportional to the number of sparse matrix multiplications performed and is given by $S * F$, where $F$ is the corresponding dense FLOPs. Thus if $F_{1\times1}(X)$ is the number of FLOPs consumed by the $1{\times}1$ convolution layers and $F(x)$ is the total number of FLOPs consumed by model $X$, we have:

$$S = (F(X) - F(Y))/F_{1\times1}(X) \tag{1}$$

Hence, sparsity measures the fraction of $1{\times}1$ convolution weights removed, and so higher sparsity indicates a smaller model. For the uniform pruning scnario, Table 1 shows the amount of sparsity

required to prune each larger FBNetV3 model to a smaller one based on Eq. (1). For global pruning, (1) does not hold, and we compute the target sparsities empirically from the layer shapes instead with details provided in Section A.2. We prune each larger FBNetV3 model to a discrete FLOPs target based on a defined set of smaller models in the family, and not to a continuous range of FLOPs values, as it makes it easier to compare models directly based on a given computation budget. If we can demonstrate that for the same computation level, the pruned larger FBNetV3 model has higher performance than a smaller model with the same FLOPs, it is sufficient to demonstrate that we can improve on the FLOPs-accuracy curve over the state-of-the-art.

## 3.2 Pruning Techniques

In this paper, we utilize a pre-trained FBNetV3 model with higher number of FLOPs without training an image classification model from scratch with sparsity, which would be time consuming and computationally intensive. There are several approaches in the literature such as prune-and-fine-tune (Han et al., 2015) and iterative pruning with sparsity scheduling (Frankle & Carbin, 2018). We have utilized the former for our experiments, as although studies have shown that iterative and incremental pruning approaches lead to better generalization performance, they typically require training for high number of epochs, need tuning and selection of optimal sparsity schedules and are computationally resource intensive. We have therefore not considered them in our experiments. For our prune and fine-tune experiments, we have used 8-GPU boxes, with each box having Nvidia V100 (Volta) 32G GPUs. As described in Section 1, we perform both global and magnitude-based pruning experiments. For the latency benchmarking, we also perform magnitude-based uniform pruning with a sparse block size of $1 \times 4$ as explained in Section 3.3.

We have conducted a hyper-parameter tuning for the learning rate parameter, with LR values in the set {4e-5, 8e-5, 1.6e-4}, as fine-tuning generally admits smaller learning rates than training from scratch. We have found that using the same learning rate for all models, along with the same hyper-parameter settings used for training the seed model is sufficient to obtain pruned networks which are superior to the baseline FBNetV3 models. Hence minimal hyper-parameter tuning was required for our experiments and we have used values of settings such as weight decay and momentum to be the same as those used for training the baseline FBNetV3 models. During fine-tuning after pruning, we have used a smoothed validation loss to stop the process early after a convergence tolerance (0.01%) is reached between two consecutive epochs. Generally, we have observed fine-tuning to converge around ∼250 epochs.

## 3.3 latency measurements and Metrics

We are interested not only in the sparsity level of our pruned models and the image recognition performance, but also in metrics which potentially improve due to model sparsity, such as number of parameters, the FLOP count and the model latency. For reporting model performance under pruning, we use standard image recognition metrics such as Top-1 and Top-5 test accuracies. We measure overall model sparsity, which is different to the layer sparsity since we only prune $1 \times 1$ convolution layers, as explained in Section 3.1. We report the model FLOPs, because this metric captures the computational footprint of the model and its power consumption.

Last, we record the total latency (in ms.) under pruning. The sparse kernels used in our experiments are already in open-source and released under the PyTorch sparse quantization library[3]. Prior to using these kernels, we perform uniform layer-wise block-based pruning with block sizes of $1 \times 4$. Magnitude based pruning is implemented at block level, and the model is quantized to 8-bit integers (int8) before latency benchmarking, which is performed on Intel CPUs designed using the Skylake micro-architecture. While we would expect sparsity to translate to tangible inference speedups, this is highly dependent on the sparse kernel support provided by hardware. Current hardware is not well-suited for unstructured randomly sparse matrix multiplications and tend to do better with structured sparsity in models (Anwar et al., 2017). We have utilized block sparsity within the weight matrix for latency experiments. However this often tends to come at a decreased level of model performance. The design of highly performant sparse models under structured sparsity with reasonable inference speedups remains an important research topic outside the scope of this paper.

---

[3]https://github.com/pytorch/pytorch/blob/master/torch/ao/nn/sparse/quantized/linear.py

Table 1: Sparsity level (in percentage) and performance of pruned FBNetV3 networks on ImageNet dataset for different target MFLOPs. The best accuracy obtained at each target FLOPs level is highlighted in bold.

| Seed network FBNetV3_ | Target network FBNetV3_ | Target MFLOPs | Baseline Accuracy | Uniform pruning | | | Global pruning | | |
|---|---|---|---|---|---|---|---|---|---|
| | | | | Sparsity level(%) | Top-1 Acc. | Gain(%) | Sparsity level(%) | Top-1 Acc. | Gain(%) |
| B | A | 356.6 | 79.6 | 26.59 | 80.308 | 0.887 | 39.5 | 80.232 | 0.793 |
| C | A | 356.6 | 79.6 | 40.7 | **80.738** | 1.43 | 57.9 | **80.476** | 1.1 |
| C | B | 461.6 | 80.2 | 19.4 | 80.996 | 0.992 | 28.9 | 80.998 | 0.985 |
| D | B | 461.6 | 80.2 | 31.47 | **81.116** | 1.142 | 43.7 | **81.08** | 1.097 |
| D | C | 557.0 | 80.8 | 15.04 | 81.278 | 0.591 | 21.5 | **81.208** | 1.256 |
| E | C | 557.0 | 80.8 | 31.0 | **81.282** | 0.596 | 43.6 | 81.184 | 0.475 |
| E | D | 644.4 | 81.0 | 17.8 | 81.118 | 0.145 | 25.8 | 81.388 | 0.479 |
| F | D | 644.4 | 81.0 | 38.2 | **82.00** | 1.234 | 67.8 | **81.484** | 0.597 |
| F | E | 762.0 | 81.3 | 29.8 | **82.19** | 1.094 | 54.7 | **81.97** | 0.824 |
| G | E | 762.0 | 81.3 | 71.67 | 81.166 | -0.16 | 85.5 | 79.934 | -1.68 |
| G | F | 1181.6 | 82.0 | 49.69 | **82.528** | 0.643 | 63.8 | **82.454** | 0.553 |

# 4    RESULTS

## 4.1    PRUNED FBNETV3 MODEL PERFORMANCE

To answer **RQ1**, we consider the family of FBNetV3 models as baselines and seed models for further pruning. For each pair of models $X$, $Y$ in the family, we calculate the amount of sparsity required to prune the larger model $X$ to a model that consumes the same number of FLOPs as the target smaller model $Y$, via Equation 1. There are 21 potential seed and target model pairs, however we conduct pruning experiments only for a depth of 2 for tractability. For example, given FBNetV3E as the seed, we only prune it to FLOPs targets corresponding to FBNetV3D and FBNetV3C. Table 1 presents the accuracy and number of parameters of the pruned models at each target FLOPs level. The improvement in performance is apparent even at lower FLOPs targets, where we might expect baseline models such as FBNetV3A to not be over-parameterized. For example, pruning FBNetV3C to a target of 356.6 MFLOPs obtains a network which is 1.43% better than FBNetV3A. Figure 1 plots the Top-1 ImageNet testing accuracy vs. FLOPs for the best pruned models as seen from Table 1. This clearly shows that pruning FBNetV3 models with minimal fine-tuning can significantly improve on the state-of-the-art for FLOPs vs. Top-1 accuracy trade-off. This analysis is performed for both uniform layer-wise and global magnitude-based prune with fine-tune settings. Global pruning ranks the weights of the entire network in contrast to uniform layer-wise pruning, which ranks each layer's weights to determine the sparsity mask. It would be expected that global pruning performs better than uniform pruning for the same target sparsity level or number of non-sparse parameters. However in our experiments we determine the pruning threshold based on FLOPs targets, and find global pruning to require higher sparsity levels, which results in uniform pruning outperforming global pruning in Top-1 ImageNet accuracy in most cases.

## 4.2    PRUNING COMPLEXITY

In addition to demonstrating the improvement over state-of-the-art obtained by pruning FBNetV3 models, it is also important to quantify the reduction in computational complexity obtained in pruning a larger FBNetV3 model compared to training an FBNetV3 model directly through NAS (Network Architecture Search). **RQ2** (pruning for efficient model search) asks if the pruning and subsequent fine-tuning approach in Section 4.1 is faster than a full-fledged neural architecture search. During pruning and subsequent fine-tuning, we train the pruned networks till the validation loss converges to within a pre-specified tolerance, as described in Section 3.2. The time needed is generally less than when training the original FBNetV3 models, which runs for 400 epochs. The number of GPU-hours is computed as (number of training GPU nodes) * (number of GPUs per node) * (training time to

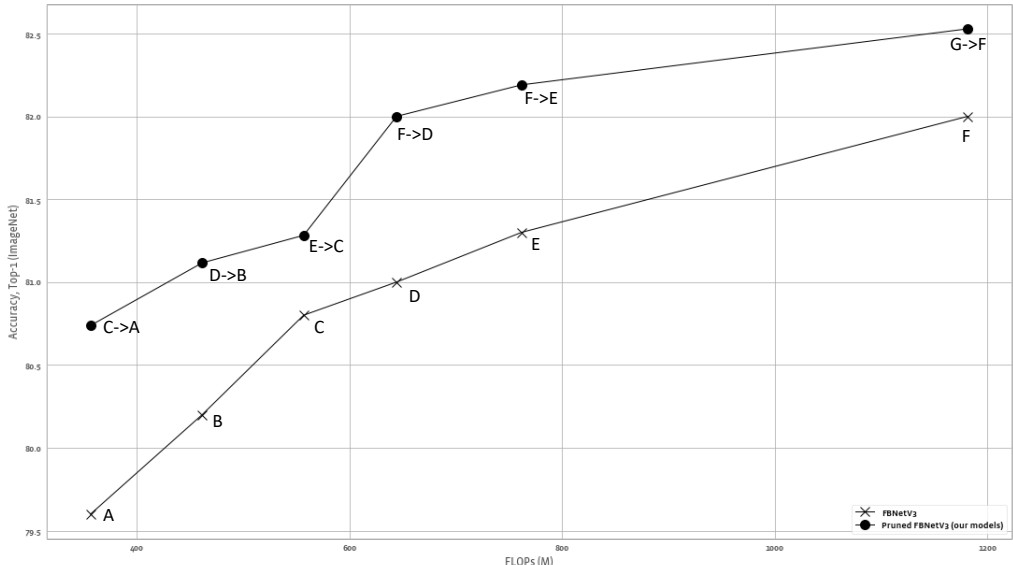

Figure 1: FLOPs vs. performance (ImageNet Top-1 acc.) for different pruned FBNetV3 networks. For comparison, the existing FBNetV3 networks are also shown here.

Table 2: Computation speedup in term of GPU-hours when comparing NAS (neural Architecture Search) with pruning and fine-tuning approaches. The selected seed networks are drawn from those in Table 1 with the best performance at target FLOPs.

| Target FLOPs (FBNetV3 Model) | GPU-hours in NAS | GPU-hours in pruning and fine-tuning | Computational cost speedup |
|---|---|---|---|
| 356.6 (FBNetV3A) | 10.7k | 2.240k | 4.77 |
| 557.0 (FBNetV3C) | 10.7k | 2.496k | 4.28 |
| 762.0 (FBNetV3E) | 10.7k | 3.456k | 3.09 |

convergence) for each network. In Table 2, for each of the best performing uniformly-pruned models in Section 4.1 we report the number of GPU-hours consumed by the prune and fine-tune strategy, along with the GPU-hours consumed when obtaining a FBNetV3 model through architecture search using the method described in Dai et al. (2020). The results are quite conclusive—we not only obtain pruned models superior in performance to the original neural search optimized models, but also as described in Section 1, computational cost is significantly lower when starting from a pre-trained model with higher FLOPs. Given the performance improvements obtained with lower computational resources, this approach is beneficial for an experimental setting where researchers have access to open-sourced pre-trained models and limited GPU resources, for example in a small startup or an academic environment. We observe that the degree of speedup reduces as the network size gets bigger (e.g., in FBNetV3A vs. FBNetV3C) due to higher training time to convergence. Nevertheless, we still obtain a speedup of 3-5 times compared to a full NAS (Neural Architecture Search).

## 4.3 LATENCY EXPERIMENTS

We also measure the latency-performance tradeoff for the pruned FBNetV3G models. FBNetV3G is the largest model in the family and so is expected to have the best generalization performance under high sparsity levels. As described in Section 3.3, we prune the network using block sparsity (where the block size is $1 \times 4$) to sparsity levels in the set {40%, 50%, 60%}. We have not utilized lower sparsity levels, as we have observed that for the selected kernels we need at least 40% sparsity to yield any significant latency benefits. We have pruned all $1 \times 1$ convolution layers uniformly here and subsequently converted them to fully-connected layers for compatibility with the quantized sparse kernels. In Figure 2a, we present the Top-1 ImageNet accuracy vs. latency curve

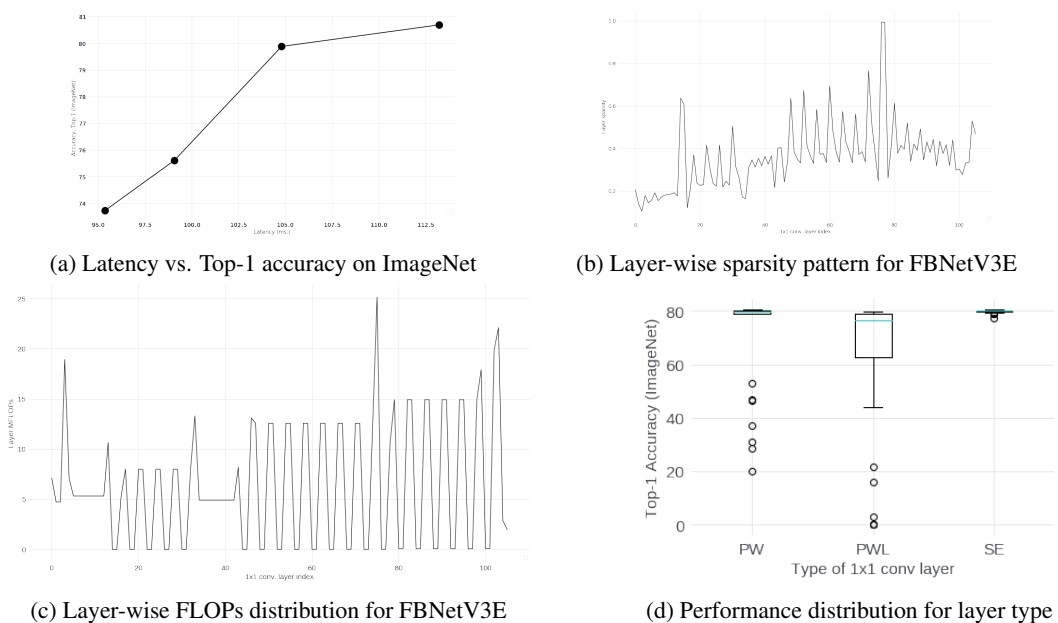

(a) Latency vs. Top-1 accuracy on ImageNet

(b) Layer-wise sparsity pattern for FBNetV3E

(c) Layer-wise FLOPs distribution for FBNetV3E

(d) Performance distribution for layer type

Figure 2: Latency benchmarking on FBNetV3G for different sparsity levels {40%, 50%, 60%} and layer-wise sparsity/FLOPs/accuracy sensitivity for a pruned FBNetV3E network.

after pruning the FBNetV3G network for different sparsity levels. The pruned FBNetV3G models show marked performance reduction with lower latency as expected, with a sparsity level of 60% translating to around 7% absolute accuracy reduction with a latency reduction of 18 ms (16% relative). While the $1\times1$ convolution layers account for >80% of FLOPs, they only constitute 25% of overall network latency. This is consistent with previous literature (Dudziak et al., 2020) which shows that computational complexity (ex. FLOPs) and latency are not well-correlated, and indeed the latter is more dependent on layer shapes. This result underscores the need to develop more latency-friendly pruning techniques which can potentially improve on the state-of-the-art in this domain.

### 4.4 INSIGHTS INTO PRUNING EXPERIMENTS

Our pruning experiments demonstrate that we can improve on the state-of-the-art FBNetV3 models in generalization performance for a given FLOPs level. In this subsection, we obtain an insight into (1) the sparsity pattern under global magnitude-based pruning and (2) the sensitivity of each layer when pruned in isolation under uniform layer-wise magnitude pruning (sparsity level of 95%). For (1) in Figure 2b, we plot the amount of sparsity obtained per $1\times1$ convolution layer. The model being considered is an FBNetV3E network pruned to a sparsity level of 43.6%, to the same FLOPs level as FBNetV3C and subsequently fine-tuned. We note that the sparsity level in lower layers is lower which is potentially required for maintaining the performance . Higher sparsity can be admitted in upper layers of the network where it has learnt more redundant representations. SE (Squeeze and Excitation) $1\times1$ convolution layers generally tend to get pruned more compared to other layers, with the sparsity being >99% for two such SE layers in stage $xif5\_0$. This indicates that we can also consider revisiting SE layer role in FBNetV3 networks, and even remove entire layers in future work to yield additional latency and FLOPs benefits.

For analysis (2) we prune each $1\times1$ convolution layer in isolation at a sparsity target of 95% and record the Top-1 test accuracy obtained on ImageNet dataset. For each type of layer, PW:expansion, PWL: bottleneck, SE: Squeeze-Excitation we plot the distribution of accuracies in Figure 2d. We observe that the PW and PWL layers are most sensitive to high sparsity, while SE layers are able to retain performance adequately. We could also avoid pruning the most sensitive layers (appearing as outliers in the figure) to maintain generalization performance. This observation corroborates findings from analysis (1), and motivates us to revisit the role of squeeze-excitation layers in future work.

## 5 CONCLUSIONS

In this paper, we have investigated the problem of improving on the current state-of-the-art FLOPs vs. performance trade-off for FBNets which have been pre-optimized by NAS (Neural Architecture Search). We have employed network pruning techniques, and our results demonstrate that we can further improve on performance over FBNetV3 at a given FLOPs target through global as well as uniform magnitude-based pruning. This happens not only for relatively over-parameterized networks such as FBNetV3G, but also smaller networks such as FBNetV3A which have lower computational complexity. On average, the GPU-hours incurred during pruning is about $\sim 4\times$ less than that consumed by a full-scale NAS. We have also performed latency measurements on the FBNetV3G model and conducted an analysis to understand the sparsity patterns and sensitivity of different FBNetV3 layers to pruning. For future work, we plan to investigate squeeze-excitation layers in more detail, and explore structured pruning approaches such as channel and layer pruning to further improve on the latency-performance tradeoff for this family of models.

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

Table 3: Baseline FBNetV3 models chosen for our experiments

| Baseline | No. of parameters (in millions) | MFLOPs | Top-1 Accuracy (ImageNet) | Top-5 Accuracy (ImageNet) |
|---|---|---|---|---|
| FBNetV3A | 8.5 | 356.6 | 79.6 | 94.7 |
| FBNetV3B | 8.5 | 461.6 | 80.2 | 94.9 |
| FBNetV3C | 9.9 | 557.0 | 80.8 | 95.3 |
| FBNetV3D | 10.2 | 644.4 | 81.0 | 95.4 |
| FBNetV3E | 10.7 | 762.0 | 81.3 | 95.5 |
| FBNetV3F | 13.8 | 1181.6 | 82.5 | 95.9 |
| FBNetV3G | 16.5 | 2129.7 | 83.2 | 96.3 |

## A APPENDIX

### A.1 DATASET

Our pruning experiments are conducted on the ImageNet dataset, which is commonly used in the literature to evaluate performance of image classification models. It is a collection of millions of images, where there is a defined taxonomy based on the WordNet hierarchy. The taxonomy comprises of approximately 22,000 visual subcategories, making this a large-scale classification problem. ImageNet was first introduced by Deng et al. (2009) and has been adopted by the machine learning and computer vision communities to benchmark image classification models. We use the entire dataset consisting of 14 million images for our experiments, and we utilize both the training set and the validation set. We split the training set to also create a smaller validation set (of 50,000 images evenly distributed across all image categories) for parameter tuning and setting the training convergence criterion. The ImageNet validation set is used in our experiments as an testing set for reporting model generalization performance.

### A.2 OBTAINING SPARSITY LEVELS FOR GLOBAL MAGNITUDE-BASED PRUNING

While the sparsity level required to prune the $1 \times 1$ convolution layers can be obtained from Equation 1 for the uniform pruning case, for global magnitude-based pruning all weights in such layers are ranked globally and then thresholded to determine the sparsity mask. There is no closed form expression we can use to determine the sparsity level given the FLOPs target for this scenario. To obtain the sparsity level, we have used the pre-trained seed model and pruned it to a sparsity level $s$. We can calculate the overall FLOPs consumed due to sparsity $s$ by plugging in the layer-wise shapes and sparsity levels. Extrapolating backwards from target FLOPs, we can easily find out which sparsity level corresponds to this. It is important to note that since we prune and fine-tune with a fixed sparsity mask, the number of FLOPs estimated at sparsity level $s$ does not change even after the network is fine-tuned.

