# OpenReview forum: "Pruning Compact ConvNets For Efficient Inference"
_ICLR.cc/2022/Conference — ICLR 2022 Submitted_

### Official Review · Reviewer_F7aS · 2021-10-28

**Correctness:** 2
**Technical Novelty And Significance:** 1
**Empirical Novelty And Significance:** 2
**Recommendation:** 3
**Confidence:** 4

**Main Review:**

The paper is very well written and clear in most aspects. Applying pruning techniques to recent architectures, especially architectures that are designed to be resource-efficient, is a task that is rarely addressed in the literature. The two main contributions as stated in the paper summary above are novel.

My main concern is about the practical relevance of the presented work. The paper is mainly an experimental contribution that shows the potential of reducing the number of FLOPS. However, reducing the number of FLOPS appears to be mostly of theoretical relevance. It is also mentioned in the paper (end of Section 4.3) that FLOPS are not well correlated with latency. I wonder if FLOPS can then be seen as a good proxy for computational footprint and power consumption as mentioned in Section 3.3. From what I know, sparsity in matrices gives a decent practical improvement only at high levels of sparsity. Special sparse-matrix structures typically store positions and values of non-zero entries and, therefore, for small amounts of sparsity the memory footprint even increases when using sparse-matrix formats. This is also supported by Section 4.3 where it is reported that sparsities below 40% do not provide a benefit in terms of latency. Consequently, I do not see a real benefit of the achieved sparsity levels in Table 1 which are often below 40%. In this view, I believe that comparing a small dense matrix with larger sparse matrices having the same number of non-zero entries is unfair. In summary, I doubt that the proposed method of pruning a larger model is "practically advantageous" as promised in the introduction.

The paper only shows experiments where pruning works reasonably well. I am missing experiments where we can see that the pruning breaks down, i.e., how far can we go until the accuracy degrades drastically on some specific model. This is important to study the impact on pruning for architectures that are inherently resource-efficient and to understand how "overparameterized" these models really are.

The second main contribution states that pruning a larger model is faster than finding a smaller model directly by means of NAS. Although the reduction in training time compared with NAS is substantial (approx. 4x), the process still seems to take around 2000--3500 GPU hours. I am not sure whether this amount of GPU hours can be considered as "can be obtained inexpensively with limited resources that are generally available to a machine learning practitioner" as mentioned in the introduction. What hardware was used for these experiments and for the latency experiments.

Minor:
It appears that F_{1x1} in Eq. (1) is the number of FLOPS of *all* 1x1 layers of the whole network. Please emphasize the "all".


**Summary Of The Paper:**

The paper presents an experimental evaluation of simple pruning techniques applied to modern architectures that are designed to be inherently resource-efficient. It is shown that pruning large models in the FBNetV3 family achieves better accuracy-FLOPS trade-offs than smaller models in the FBNetV3 family. It is also shown that pruning large models is faster than performing NAS to find the smaller models directly.

**Summary Of The Review:**

The paper is largely experimental and, in my opinion, shows improvements in FLOPS that are not practically relevant. However, the paper promises a few times that the proposed methods provide practical benefits. I am not sure, what to take away of the paper: in my opinion there are hardly any practical benefits and there is no clear takeaway message from the theoretically relevant decrease in FLOPS.

---

> ### Author Response · Authors · 2021-11-22
> **Rebuttal to Reviewer F7aS**
>
> Regarding practical relevance of the proposed work, the objective of this paper is to investigate whether pruning can improve on state-of-the-art FBNetV3 models to obtain better generalization performance at the same FLOPs level. Even though we measure latency as a side-experiment, we do not have any research questions around it. For the purpose of this work, we consider FLOPs as an acceptable metric for demonstrating practical utility of our models as :
> 1. FLOPs correlates with power consumption and memory overhead ex. paper [1]
> 2. Current literature ex. papers [2] demonstrate improvement on model FLOPs and from a research standpoint, they have made meaningful contributions.
> While lower sparsity levels do not translate to tangible latency reductions, some of the models we have trained actually demonstrate latency reductions at sufficiently high sparsity levels.
>
> To put the amount of computation in perspective, consider the case of an AWS environment with a single GPU box (it is standard to have 8 GPUs in each box, and training is distributed which we have done here). For p3.16xlarge,  3000 gpu-hours = 3000/8 instance hours = approx. 3000 dollars [3]. To reach a specified number of FLOPs, where the target is determined based on the use-case, we have shown that the cost incurred is around 4x less.  A key assumption here is that the practitioner can afford to train a FBNetv3 model at least once. If that is the case, a practitioner training an FBNetV3 model incurs no additional cost to prune and re-train a pre-trained model, while NAS with the same objective would cost ~12,000 dollars. This would mean an overall saving of ~9000 dollars.
>
> For these experiments, we have used 8-GPU boxes, with each box having Nvidia V100 (Volta) 32G GPUs.
>
> References
> [1] Tang, Raphael et al. “An Experimental Analysis of the Power Consumption of Convolutional Neural Networks for Keyword Spotting.” 2018 IEEE International Conference on Acoustics, Speech and Signal Processing (ICASSP) (2018): 5479-5483.
> [2] Molchanov et al., Pruning Convolutional Neural Networks for Resource Efficient Inference, ICLR 2017
> [3] https://aws.amazon.com/ec2/pricing/on-demand/

---

> > ### Comment · Reviewer_F7aS · 2021-11-29
> > **Response**
> >
> > I thank the authors for their response. It is obvious that using fewer GPU hours directly translates into a lower cost. I am still wondering if what we obtain after the shorter time (a large but sparse model) is preferable to what we obtain after the longer time (a small but dense model). I am also inclined with the other reviewers that the paper would benefit if more general insights beyond the evaluated architectures and pruning techniques would be provided.

---

### Official Review · Reviewer_tgou · 2021-11-01

**Correctness:** 3
**Technical Novelty And Significance:** 1
**Empirical Novelty And Significance:** Not applicable
**Recommendation:** 3
**Confidence:** 4

**Main Review:**

Pros:
1) The paper is generally well-written and easy to follow.
2) The authors provide valuable evaluations to show that pruning can be useful even for optimized network architectures, i.e. those searched by NAS.

Cons:
1) The technical contribution and novelty are limited. This paper only applied previous pruning methods to the network obtained by NAS and made a series of evaluations and comparisons. Technically, this paper didn't make novel contributions. And considering the experimental evaluations, the conclusions are not surprising to me, i.e. they didn't indicate or inspire new scientific problems. So generally, I think the contributions are quite limited.

2) There are many typos in Table 1. The gains shown in Table 1 seem incorrect. It is strange that the shown numbers are not equal to (accuracy of pruned models - accuracy of baselines). And in the last line of Table 1, the baseline accuracy seems incorrect, i.e. it should be 82.5.

3) Some details are not clear. For example, in Fig. 1, what are the types of baseline models and corresponding seed networks? For Table 2, what are the selected seed networks? Are the selected seed networks the same in Table 2?

4) The evaluation is only based on FBNetV3 architecture. I just doubt the generality of the conclusions.

**Summary Of The Paper:**

This paper applies conventional pruning-and-finetuning techniques to further compress the networks searched by NAS. The experiments and evaluations are based on the family of FBNetV3. The authors show that by pruning large FBNetV3 model to small one, the accuracy of pruned model may be slightly better than the original target (small) model, achieving better tradeoff between computational complexity and accuracy. Also, the authors show that pruning is more training-efficient than searching by NAS to achieve a compact FBNetV3 model.

**Summary Of The Review:**

I made my recommendations mainly considering the limited contribution and limited practical significance made by this paper.

---

> ### Author Response · Authors · 2021-11-20
> **Response to Reviewer tgou**
>
> In this paper, the technique we have employed for pruning (unstructured pruning) is already tried and tested, however we are not claiming novelty for the approach. Our novelty lies in studying whether efficient image recognition models such as FBNetV3 can be optimized further to improve on the FLOPs-accuracy curve, and the contributions are two fold : (1) FBNets are themselves state-of-the-art in efficient vision models and we achieve better accuracy-FLOPs tradeoff over the state-of-the-art and (2) from the standpoint of computational overhead, we significantly reduce the amount of GPU hours required to obtain such models. Pruning a publicly available NAS optimized model incurs ~4x less GPU hours to achieve a target FLOPs level, compared to training a full-fledged NAS to obtain a model which has less accuracy at the same FLOPs level.
>
> We did not identify many typos in Table 1. As mentioned in table, we have reported % gains in accuracy (instead of absolutes), so it would not match the absolute difference. The last line of Table  has the FLOPs corresponding to the target model (not the source model), so it is correct for the FBNetV3F model. The Figure 1 shows higher performance for the FBNetV3F model, so we have corrected that.
>
>  We will revise Fig 1 to indicate that each seed network and pruned model is a member of FBNetV3 family (A, B, C,...G) to make it clearer. The selected seed networks in Table 2 are a subset of the ones in Table 1, we will revise to make this clearer.
>
> In theory, we could replicate these experiments on other efficient networks such as EfficientNet and MobileNets. However, the FBNetV3 models are already shown to be higher-performant at a FLOPs level compared to these models which still have some level of over-parametrization.  Hence, it is expected that any pruning experiments would yield the same results. Consequently we have reported results only for the FBNetV3 model and mentioned our reasoning in Section 2 (Related Work/Computationally Efficient Vision Models).

---

### Official Review · Reviewer_haMF · 2021-11-02

**Correctness:** 1
**Technical Novelty And Significance:** 1
**Empirical Novelty And Significance:** 1
**Recommendation:** 1
**Confidence:** 4

**Main Review:**

Paper is primary focused on empirical results. Applied technique, magnitude pruning, is a well known techniques.

Two research questions that paper tries to answer are interesting, but have been answered multiple times.

It is known that pruning a larger model to the level of smaller will give a more accurate model. Researchers have been showing this by pruning Resnets from 101 to 50 to 34 to 18 etc.

Pruning is more efficient to get a model derivative than running a new NAS targeted to a smaller model, also models like EfficientNet provide a scaling rule that doesn't require to run NAS again for a different computational budget.

For an overview of recent work in pruning please refer to [R1]. MobileNet and EfficientNet are popular model to benchmark a pruning technique. For this kind of study, it is highly recommended to apply multiple pruning techniques to get technique agnostic insights.

R1) Davis W. Blalock, Jose Javier Gonzalez Ortiz, Jonathan Frankle, John V. Guttag, What is the State of Neural Network Pruning? MLSys 2020

**Summary Of The Paper:**

Paper focuses on pruning 1x1 convolutions in FBNet with existing technique (magnitude pruning). Authors tend to answer 2 questions: a) how a larger pruned model compare to smaller model trained from scratch; b) is pruning an efficient way to get NAS model derivatives.

**Summary Of The Review:**

Paper presents an empirical study on a well studied problem. The setup, method and results are not significant and are well known to the community.

---

> ### Author Response · Authors · 2021-11-20
> **Response to Reviewer haMF**
>
> Thank you for your feedback on the paper. We will be citing the mentioned paper by Frankle et al. in a revised draft.
>
> The main concern in the feedback is that the findings are not novel ie. with other networks it has been shown that pruning a larger model to the level of a smaller model provides higher accuracy. While this is true for many networks, it is not guaranteed this will happen for every pair of larger and smaller models. If the larger model is not highly over-parameterized it would not be expected that pruning it will always have higher performance than a NAS optimized smaller model. Particularly this open question has not been investigated for FBNetV3 models in previous literature, even though studies have been done for networks such as ResNet, EfficientNet and MobileNet. For example, in our paper we find that while most pruned larger FBNetV3 models do better than smaller models and improve FLOPs-accuracy curve, in certain scenarios this does not happen eg. when pruned G does not perform better than E.

---

> > ### Comment · Reviewer_haMF · 2021-11-24
> > **Thanks for response**
> >
> > Thank you for providing a rebuttal. In my opinion, applying a single existing technique to a single model architecture is not significant. Moreover, getting exactly the same findings as for previous architectures makes the contribution even lower. I encourage authors to apply more pruning techniques, evaluate on multiple model architectures and find common insights that do not exist in the previous works.

---

### Official Review · Reviewer_gdjt · 2021-11-03

**Correctness:** 4
**Technical Novelty And Significance:** 1
**Empirical Novelty And Significance:** 3
**Recommendation:** 3
**Confidence:** 4

**Main Review:**

Strengths
- This paper is the among the first if not the first to demonstrate that FBNetv3 (a state-of-the-art compact network architecture searched with a sophisticated NAS method) can be pruned to achieve better computation-accuracy performance, than searching for the architecture at the same complexity. It also shows that pruning the larger FBNetv3 model gets to a higher accuracy more quickly than running architecture search.
- The experimental results are good and they show that at different FLOP points, pruned FBNetv3 models always perform better than their original searched counterparts.

Weaknesses
- The paper has limited novelty and does not present any ideas on its own in terms of the pruning method. It merely uses some existing pruning methods on the FBNetv3 architecture in a straightforward manner.
- It fails to make experimental comparison to NAS methods that jointly search and prune, such as the cited paper `(Wang et al., 2020)`.
- Only FBNetv3 is evaluated in the paper. How about other compact NAS architectures? It seems like the authors tried with other architectures but find that they do not work well with pruning. It could be that the claims in the paper are only narrowly applicable to FBNetv3.


**Summary Of The Paper:**

This paper describes the details on how the authors prune FBNetV3 (which is already a compact network architecture) models with existing methods to make them more compact. The resulting models have better FLOP-accuracy tradeoffs than the original FBNetV3.

**Summary Of The Review:**

This paper has some good empirical contributions but those are all it has to offer. For a paper like this, I wish to see a stronger version of the paper where pruning is generally applicable to all sorts of compact NAS architectures.

---

> ### Author Response · Authors · 2021-11-20
> **Response to Reviewer gdjt**
>
> Thank you for your feedback on the paper.
>
> While the pruning method used in our experiments is not new we have 1. formulated a hypothesis which solves a novel problem of making NAS-optimized networks more efficient (this is an insufficiently studied domain) and 2. our results are novel - pruning can improve on state-of-the-art FBNetV3 models in terms of the performance vs. FLOPs tradeoff, which has not been done before. We do not claimed that our pruning techniques are novel, and will modify the paper to clarify this.
>
> For a direct evaluation of the effectiveness of our experiments (ie. based on the hypothesis that we can improve on the tradeoff between FLOPs and performance), we consider FBNetV3 as a baseline, as it is state-of-the-art, and we optimize  those models further. Regarding other techniques, we do not compare with them due to experimental settings and metrics in those papers being very different from ours. For example the paper we have cited (Wang, 2020) solves a different problem than improving on performance vs. FLOPs: they consider current search of pruning policy within NAS as insufficient, and also incorporate search for a quantization policy into their approach which we do not consider in our main experiments (we only quantize during latency reporting as required by the sparse kernel we have used). They also report BitOps as a metric, as FLOPs doesn't make sense for quantized models. Another bottleneck is the type of resources for benchmarking - the paper uses BitFusion, while we report metrics on commodity hardware such as Intel Skylake CPUs.
>
> While it is possible to repeat our experiments on other networks such as EfficientNet and MobileNets, the authors of the FBNetV3 paper have already done a study and shown that the  FBNetV3 models can achieve better accuracy on ImageNet at a given FLOPs level compared to these models. This indicates that these networks are over-parametrized and ripe for pruning experiments, thus we would expect the findings from FBNetV3 to translate to these architectures. Consequently we have reported results only for the FBNetV3 model and mentioned our reason in Section 2 (Related Work/Computationally Efficient Vision Models). We will modify the paper to better explain our reasoning for conducting experiments only on FBNetV3.

---

### Decision · Program_Chairs · 2022-01-20

**Decision:**

Reject

**Comment:**

This paper presents an empirical study which shows that pruning FBNets with larger capacity results in a model with higher accuracy than one searched via neural architecture search. The below are pros and cons of the paper mentioned by the reviewers:

Pros
- The observation that optimized architectures such as FBNets can benefit from pruning is interesting.
- The paper is well-written and easy to follow.

Cons
- It is trivially known that training larger model and then pruning it will yield a better performing model, than training a smaller model from scratch.
- The authors do not propose a novel pruning technique for optimized CNN architectures, and use existing pruning techniques for all experiments.
- The experimental validation is only done with FBNets on ImageNet, and it does not show when pruning starts to break down.

All reviewers unanimously voted for rejection, especially since the main “findings” of this paper that compact architectures can be further pruned down for improved accuracy/efficiency tradeoff, and that pruning a larger compact model results in models that outperform smaller models trained from scratch, have been already shown in many of the previous works on neural pruning. In fact, compact networks such as MobileNets and EfficientNets are the standard architectures for measuring the effectiveness of pruning techniques, and thus the contribution of this work reduces down to showing that the same results can be obtained with FBNets. This could be of interest to some practitioners, but is definitely not sufficient to warrant publication.